# Text-Guided 3D Head Synthesis Using Geometry Images

## Abstract

In recent years, text-guided 3D head generation has advanced considerably with the development of 3D morphable models (3DMMs) and their integration with vision–language models (VLMs). Nevertheless, existing approaches remain limited by the coarse level of detail in commonly used 3DMMs, which restricts their ability to synthesize fine-grained facial geometry and complex expressions. To address this limitation, we propose a novel framework for text-guided expressive 3D head generation. Unlike prior works that directly operate on mesh-based representations, our method leverages geometry images as the core 3D shape representation. Our method begins by computing a measure-preserving parameterization for each head mesh, minimizing area distortion while allowing local magnification of regions of interest. This parameterization enables the construction of geometry images, which we then use to train a conditional Denoising Diffusion Probabilistic Model (DDPM). By reformulating 3D generation as a 2D image synthesis problem, our framework excels at capturing fine-grained geometric details and expressive deformations that mesh-based pipelines often fail to reproduce. Extensive quantitative and qualitative experiments demonstrate that our approach produces high-quality human avatars and consistently outperforms existing methods.

## 1 Introduction

3D head modeling and generation have become increasingly important across industries such as virtual and augmented reality (VR/AR), film production, and gaming. Traditional approaches typically depend on manual sculpting, which requires skilled artists and is both time-consuming and labor-intensive. A major breakthrough came with the introduction of the first 3D Morphable Model (3DMM) by Blanz & Vetter (1999), which is a statistical model for deforming a template mesh through parameter adjustments to achieve accurate 3D head fitting. Building on this foundation, numerous 3DMM-based methods have been developed for 3D head synthesis (Cao et al., 2013; Li et al., 2017; Sanyal et al., 2019). With the rapid advancement of vision–language models, text-guided 3D generation has emerged as a prominent research direction, allowing the synthesis process to be explicitly controlled by natural-language prompts (Poole et al., 2022; Lorraine et al., 2023; Liang et al., 2024). Within this domain, considerable attention has been devoted to text-guided 3D head and face generation or editing (Wu et al., 2023; 2024; Zhang et al., 2023; Liao et al., 2024). These methods typically employ neural networks to adapt 3DMM templates and to generate corresponding textures conditioned on textual input.

However, the heavy reliance of these methods on 3DMM templates fundamentally limits the quality of the generated faces. Unlike high-resolution head scans obtained through 3D scanning, which typically contain 400k to over one million vertices, 3DMMs generally consist of fewer than 40,000 vertices. This coarse resolution hinders the ability to capture fine-grained geometric details and to represent complex facial expressions. There are also works (Chen et al., 2024; Luo et al., 2025) generate high-quality 3D face using neural implicit representations, such as NeRF (Mildenhall et al., 2021) and Gaussian Splatting (Kerbl et al., 2023). However, they require additional algorithms, such as Marching Cube (Lorensen & Cline, 1998), to convert the generated results into triangular meshes in order to fit common computer graphics pipeline, which typically results in significant time consumption and loss of precision.

Recently, Geometry image (Gu et al., 2002; Sander et al., 2003), as a 3D representation, is increasingly being used in 3D generation tasks (Alhaija et al., 2022; Elizarov et al., 2025; Yan et al., 2025). First introduced by Gu et al. (2002), this technique encodes 3D surfaces into 2D image formats, allowing 3D data to be processed within the 2D domain and can save huge memory usage and computational cost. Later, Sander et al. (2003) proposed multi-chart geometry image to handle surfaces with high genus or multiple boundaries by partitioning the 3D surface into smaller patches. This approach has since been adopted for 3D object generation under class- or text-conditioned settings (Elizarov et al., 2025; Yan et al., 2025). However, a key drawback of these methods is that the generated 3D results often exhibit noticeable cracks, since the correspondence across patch boundaries is discarded when generating geometry images. Additional stitching algorithms are required during 3D reconstruction, which incurs extra computational time and does not guarantee good quality. Moreover, these methods are not tailored for 3D face or head generation, and their reliance on multi-chart geometry images often leads to suboptimal results or even complete failure when applied to facial data (Elizarov et al., 2025).

In this paper, we present a novel framework for text-guided expressive 3D head generation using geometry images as the core 3D shape representation. Instead of directly operating on raw 3D mesh data as in prior works, we reformulate the problem by training a text-guided Denoising Diffusion Probabilistic Model (DDPM) on geometry images. The model samples geometry images, which are then reconstructed into high-quality textured meshes with expressions faithfully aligned to the input text prompts. Our main contributions are as follows:

- To the best of our knowledge, we are the first to apply measure-preserving geometry images as the 3D representation in text-guided expressive face generation.
- We train a text-guided conditional Denoising Diffusion Probabilistic Model for geometry image generation to obtain 3D human heads by generating 2D images.
- We demonstrate the superior performance of our framework through extensive qualitative and quantitative experiments, consistently surpassing prior 3DMM-based methods in terms of geometric detail, expressiveness, and realism.

The advantages of our methods are the following.

- **Dimension reduction**. Applying 2D geometry images as 3D representations reduces storage usage and related computational cost, allowing generated **high-resolution meshes** with more vertices and triangles.
- **Excellent details and facial expressions**. The measure-preserving optimal transport ensures that geometry images retain more facial features, resulting in generated 3D heads with more realistic details and expressions.
- **Texture and and texture mapping**. Texture images are generated alone with geometry images and texture mapping is also inherently provided.
- **Compatibility**. It is easy to apply post-processing algorithms to the generated results, such as Laplacian smoothing (Vollmer et al., 1999). Moreover, the generated results can be easily compatible with common computer graphics pipelines and rendering techniques such as normal mapping (Cohen et al., 1998).

## 2 METHOD

In this section, we elaborate the detailed processes of the proposed OT-based geometry image generative framework, which includes computing parameterization, generating geometry images, training the neural network, 3D reconstruction and post-processing. The pipeline is shown in Figure 1.

### 2.1 INITIAL PARAMETERIZATION

Since a 3D human head can be represented as an oriented surface with zero genus and one boundary (i.e., a topological disk), there always exists a homeomorphism (global parameterization) between such a surface and a planar square. Consequently, a single global parameterization is sufficient for our task, avoiding the need for multi-chart geometry images that often introduce cracks and seams.

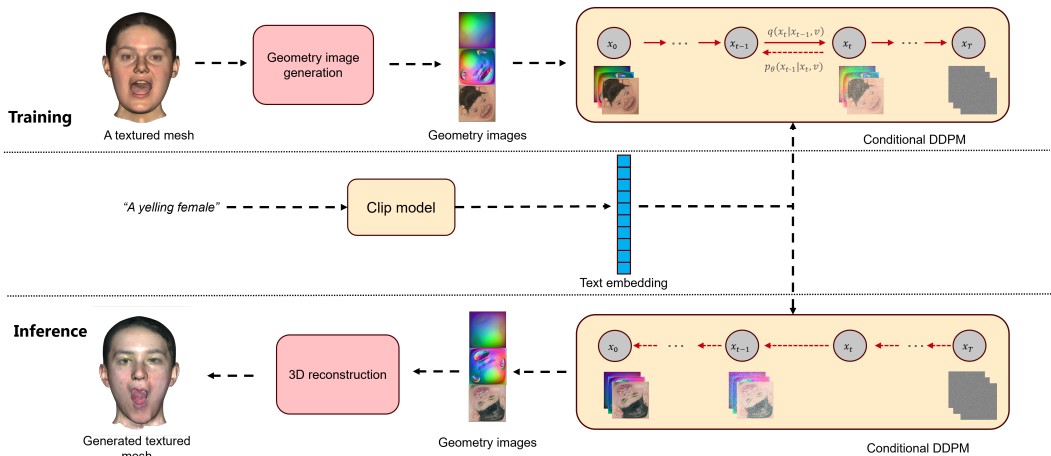

Figure 1: **Pipeline of the proposed generative framework.** Textured mesh data are converted to geometry images using measure-preserving parameterization. During training, geometry images and corresponding text embedding are fed into the DDPM. During inference, the model takes a text prompt describing the expression and gender, and samples geometry images which match the prompt. The sampled images can be further reconstructed into 3D head meshes with textures.

We first apply the Ricci Flow algorithm (Jin et al., 2008) to compute an initial conformal parameterization so the 3D surface is mapped to a 2D rectangular domain. Ricci Flow is a powerful tool to compute a metric on a surface which satisfies any given target curvatures, as long as the curvatures meet the *Gauss-Bonnet Theorem*. For any textured mesh of a human head, which is a surface with zero genus and one boundary, we select four vertices on the boundary as the corner points and set their target curvatures to $\frac{\pi}{4}$, and zero elsewhere. The objective is to optimize the Ricci energy

$$E(\mathbf{u}) = \int_{\mathbf{0}}^{\mathbf{u}} \sum_i (\bar{K}_i - K_i) du_i, \tag{1}$$

where $\mathbf{u}, \bar{K}$ and $K$ are the *conformal factor*, target and current Gaussian curvature, respectively. Such energy can be optimized in Newton's method, by calculating the gradient $(\bar{K}_i - K_i)^T$, and Hessian

$$\begin{cases} \frac{\partial K_i}{\partial u_i} = -\sum_j w_{ij} & \text{on the diagonal,} \\ \frac{\partial K_i}{\partial u_j} = \frac{\partial K_j}{\partial u_i} = w_{ij} & \text{elsewhere.} \end{cases} \tag{2}$$

where $w_{ij}$ are the cotangent edge weights. After the optimization converges, we embed all edges and obtain a conformal parameterization of the 3D mesh on the 2D domain, which is a rectangle with four selected vertices at the corners. We refer readers to Jin et al. (2008) for more details of the theory and computation.

## 2.2 MEASURE-PRESERVING PARAMETERIZATION

A conformal parameterization preserves angles locally but will inevitably introduce large area distortions. Regions away from the boundary may be densely clustered in some small areas of the 2D parameterization domain, which causes uneven sample density and detail loss during the reconstruction, as shown in Figure 2. To address this issue, we followed the optimal transport algorithm (Zhao et al., 2013) to generate a measure-preserving parameterization from the initial conformal parameterization.

We briefly review the computation method of optimal transport. The optimal transport problem was originally proposed by Monge (1781).

**Problem 2.1 (Monge's Problem)** *Suppose $(X, \mu), (Y, \nu)$ are metric space with measures that satisfy $\int_X \mu dx = \int_Y \nu dy$. We say a map $T : X \to Y$ is measure preserving if for any measurable*

set $B \subset Y$, $\mu(T^{-1}(B)) = \nu(B)$. *Given a transportation cost function* $c : X \times Y \to \mathbb{R}$, *find the measure preserving map* $T$ *that minimizes the cost* $C(T) := \int_X c(x, T(x))d\mu(x)$.

In the discrete setting, suppose $\mu$ has compact support on $X$, $\Omega = supp\, \mu = \{x \in X | \mu(x) > 0\}$ and $\Omega$ is convex. Let Y be a discretized point set $\{y_1, y_2, \ldots, y_k\}$ with Dirac measure $\nu = \sum_{j=1}^{k} \nu_j \delta(y - y_j)$. We define a *height vector* $\mathbf{h} = (h_1, h_2, \ldots, h_k) \in \mathbb{R}^k$. For $\forall y_k \in Y$, we construct a hyperplane on $X$, $\pi_i(\mathbf{h}) : \langle x, y_i \rangle + h_i = 0$. Then, function $u_{\mathbf{h}}(x) = \max_{i=1}^{k}\{\langle x, y_i \rangle + h_i\}$ is a convex function. Its graph $G(\mathbf{h})$ is an infinite convex polyhedron with supporting planes $\pi_i(\mathbf{h})$. Then we can obtain a polygonal partition of $\Omega$ from the projection of $G(\mathbf{h})$, which is equivalent to a power diagram $D(\mathbf{h})$,

$$\Omega = \cup_{i=1}^{k} W_i(\mathbf{h}) \tag{3}$$

$$W_i(\mathbf{h}) = \{x \in X | u_{\mathbf{h}}(x) = \langle x, y_i \rangle + h_i\} \cap \Omega. \tag{4}$$

$W_i(\mathbf{h})$ is the projection of a facet of $G(\mathbf{h})$ onto $\Omega$ and its area is given by

$$w_i(\mathbf{h}) = \int_{W_i(\mathbf{h})} \mu(x)dx. \tag{5}$$

Since the convex function $u_{\mathbf{h}}$ on each cell $W_i(\mathbf{h})$ is a linear function $\pi_i(\mathbf{h})$, the gradient map, $grad\, u_{\mathbf{h}} : W_i(\mathbf{h}) \to y_i, i = 1, 2, \ldots, k$, maps $W_i(\mathbf{h})$ to a single point $y_i$. We Define the admissible space of the height vectors $H_0 := \left\{ \mathbf{h} | \int_{W_i(\mathbf{h})} \mu > 0, \sum_i h_i = 0 \right\}$. Then, define the energy $E(\mathbf{h})$ as the volume of the convex polyhedron bounded by the graph $G(\mathbf{h})$ and the cylinder through $\Omega$ minus a linear term:

$$E(h) = \int_{\Omega} u_{\mathbf{h}}(x)\mu(x)dx - \sum_{i=1}^{k} \nu_i h_i. \tag{6}$$

The gradient of the energy is given by

$$\nabla E(\mathbf{h}) = (w_1(\mathbf{h}) - \nu_1, w_2(\mathbf{h}) - \nu_2, \ldots, w_k(\mathbf{h}) - \nu_k)^T. \tag{7}$$

Suppose the cells $W_i(\mathbf{h})$ and $W_j(\mathbf{h})$ intersect at edge $e_{ij} = W_i(\mathbf{h}) \cap W_j(\mathbf{h}) \cap \Omega$, the the Hessian of $E(\mathbf{h})$ is given by

$$\frac{\partial^2 E(\mathbf{h})}{\partial h_i \partial h_j} = \begin{cases} \frac{1}{|y_j - y_i|} \int_{e_{ij}} \mu, & W_i(\mathbf{h}) \cap W_j(\mathbf{h}) \cap \Omega, \\ 0, & \text{elsewhere.} \end{cases} \tag{8}$$

Suppose the conformal parameterization obtained from Ricci flow is denoted as $\phi$. Our goal is to find a map $\psi$ such that a measure-preserving mapping is given by $\psi^{-1} \circ \phi$. The detailed pipeline is shown in Figure 1. Suppose the surface $(S, \mathbf{g})$ is represented by a triangle mesh $M(V, E, F)$. For any $v_i \in V, i = 1, 2, \ldots, k$, let $p_i = \phi(v_i), p_i \in P$. For each $p_i$, we define the discrete measure $\nu_i$ given by $\nu_i = \frac{1}{3} \sum_{[v_i, v_j, v_k] \in F} \lambda_i \cdot Area([v_i, v_j, v_k])$, where $[v_i, v_j, v_k]$ is a face adjacent to $v_i$ and $\lambda_i \in [1, +\infty)$ is the scaling factor. The summation of the measure is normalized so that $\sum_i \nu_i = 1$. We also define a 2D square $\Omega$ with measure $\mu$ such that $P$ is inside $\Omega$ which should be the region of the final parameterization. The scaling factor $\lambda$ can modify the original area measure, allowing the region of interests (ROI) to be magnified, so that more points inside ROI can be sampled, as shown in Figure 2. For our task, we are more interested in the facial area than in other areas such as the back of the head and neck. Thus, we set $\lambda_i$ be 2.5 on facial region and 1 elsewhere.

The computational procedures are shown in Algorithm 1 in Appendix A.1. Initially, we set the *height vector* $\mathbf{h} = (|p_1|, |p_2|, \ldots, |p_k|)^T$. Then we compute the power diagram $D(\mathbf{h})$ and Delaunay triangulation $T(\mathbf{h})$. After projection and calculating the cell areas $\mathbf{w}(h)$ by eq. (5), we can use Newton's method to optimize the energy eq. (6). During the iterations, the *height vector* is required to be in the admissible space, so that each cell $W_i(\mathbf{h})$ is non-empty. If there is an empty cell, we roll back the calculation, half the step length and recompute $\mathbf{h}$ until there is no empty cell. Finally we have $\psi : \Omega \to P, W_i(\mathbf{h}) \to p_i, i = 1, 2, \ldots, k$.

|  | **Position** | **Normal** | **Texture** | **Reconstruction** |
|---|---|---|---|---|

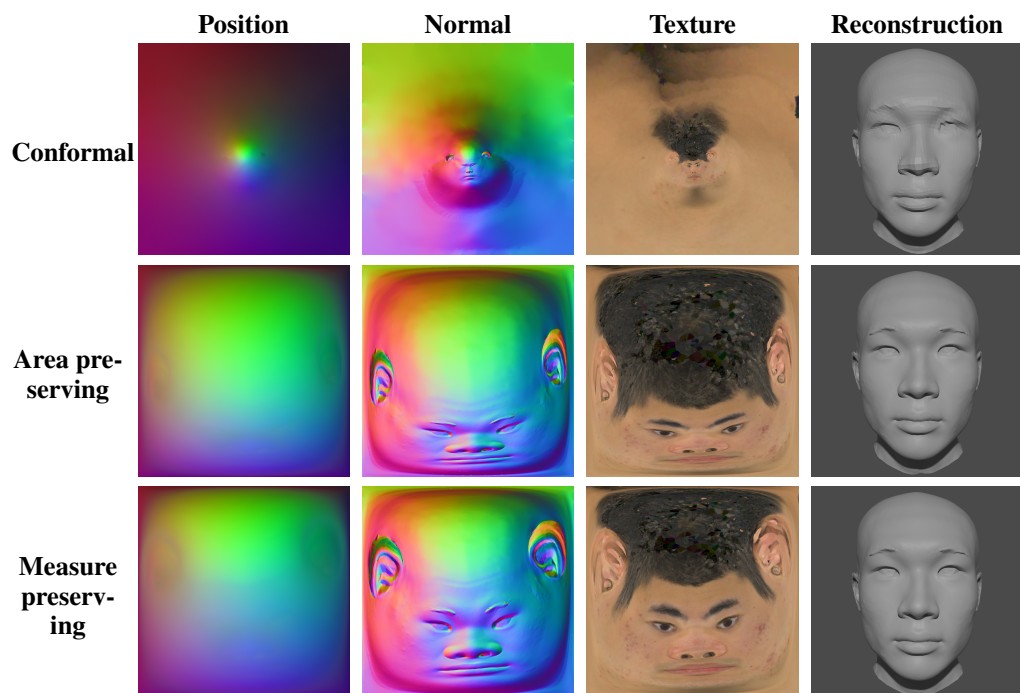

Figure 2: Comparison of geometry images and reconstructed meshes using different parameterizations: conformal, area-preserving and measure-preserving parameterization. measure-preserving parameterization is obtained by setting scaling factor $\lambda = 2.5$. Please zoom in on the images to discern the differences in reconstruction quality.

### 2.3 GEOMETRY IMAGE GENERATION

After we obtain the parameterization for each human head, we can generate geometry images including position and normal. This can be achieved by setting the normalized xyz coordinates and normal direction to each vertex as RGB values on the planar mesh and rendering the images, respectively. We also render the new texture images according to the measure-preserving parameterization. All images are stored in 16-bit PNG format for better precision.

### 2.4 DDPM FOR GENERATION

We train a conditional DDPM (Ho et al., 2020) as the text-guided generative model. The input data are the channel-wise stacking of the position, normal and texture images with a size of $9 \times 512 \times 512$, and a text prompt describing the gender and expression of the identity. The text prompt is passed through a pre-trained CLIP model (Radford et al., 2021) to obtain an embedding vector, which is later fused with image features through the cross-attention (Vaswani et al., 2017) mechanism in the down-sampling blocks, up-sampling blocks and middle layers of the UNet (Ronneberger et al., 2015) backbone, as shown in Figure 3.

For the forward process of DDPM, we add noise following the glide cosine scheduler (squared-cos_cap_v2) provided by HuggingFace Diffusers (von Platen et al., 2022). The forward process reads

$$x_t = \sqrt{\bar{\alpha}_t}x_0 + \sqrt{1 - \bar{\alpha}_t}\epsilon, \epsilon \sim N(0, I) \tag{9}$$

and the backward process is

$$x_{t-1} = \mu_\theta(x_t, t, v) + \sigma_t z, z \sim N(0, I) \tag{10}$$

$$\mu_\theta(x_t, t, v) = \frac{1}{\sqrt{\alpha_t}}(x_t - \frac{1 - \alpha_t}{\sqrt{1 - \bar{\alpha}_t}}\epsilon_\theta(x_t, t, v)) \tag{11}$$

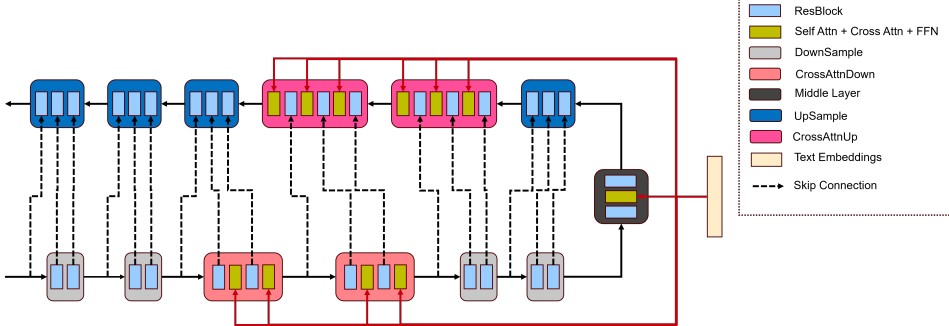

Figure 3: Demonstration of the network used for DDPM.

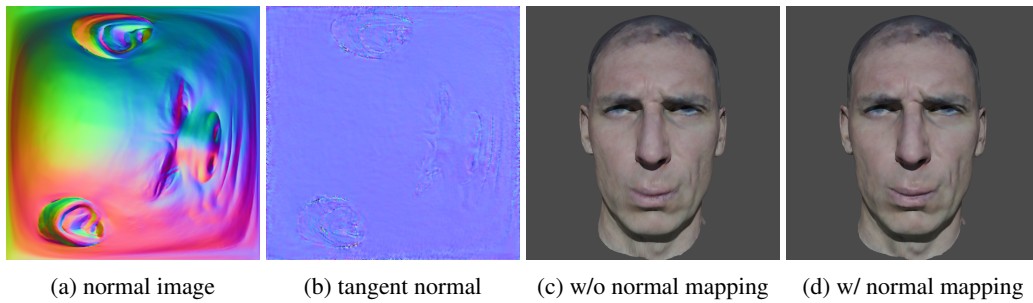

(a) normal image      (b) tangent normal      (c) w/o normal mapping      (d) w/ normal mapping

Figure 4: Generation and application of normal map, which can relieve the high anisotropy or artifacts introduced by Measure-preserving parameterization and improve the details.

where $v$ is the text embedding. We predict the noise added to the images in three different modalities separately and the objective function is

$$L(\theta) = \mathbb{E}_{(x_0,v),\epsilon,t}[||\epsilon^{pos} - \epsilon_\theta(x_t^{pos}, t, v)||_1 + ||\epsilon^n - \epsilon_\theta(x_t^n, t, v)||_1 + ||\epsilon^{tex} - \epsilon_\theta(x_t^{tex}, t, v)||_1] \quad (12)$$

We utilize AdamW optimizer (Loshchilov & Hutter, 2017) with an initial learning rate of 1e-5 to train the model for 900k steps.

We also experimented with Latent Diffusion Models (LDMs) (Rombach et al., 2022), following the approach of Elizarov et al. (2025) for geometry image generation. In this setup, images are first encoded into a latent space using a VAE (Kingma & Welling, 2014), and the diffusion process is then carried out in that latent space. However, existing pre-trained VAEs perform poorly on geometry images, as geometric data requires much higher reconstruction precision than natural images. While the decoded images may appear visually flawless, the corresponding reconstructed meshes often contain severe noise and distortions. To address this issue, we further attempted to fine-tune the VAE on our geometry image dataset, as suggested in Elizarov et al. (2025). Nevertheless, we consistently struggled to balance the KL divergence loss with reconstruction fidelity, preventing the VAE from providing a reliable latent space and ultimately undermining the generative process. We speculate that this difficulty arises in part from the limited size and diversity of our dataset compared to the one in Elizarov et al. (2025).

## 2.5 3D RECONSTRUCTION AND POST-PROCESSING

The generated geometry images can be directly converted back into 3D head models. Specifically, by mapping the RGB values of each pixel in the position image to the $x, y, z$ domain, every pixel corresponds to a vertex of the reconstructed mesh. The connectivity is then recovered by linking the diagonals of the quadrilaterals formed by four neighboring pixels, yielding the edges and triangular faces of the mesh. Moreover, since geometry images inherently encode surface parameterization, with UV coordinates naturally aligned to normalized pixel coordinates, the reconstruction process also enables straightforward generation of textured meshes.

It is important to note that measure-preserving parameterization inevitably introduces angle distortions, which may cause the reconstructed surfaces to appear flawed, visually. To mitigate this issue,

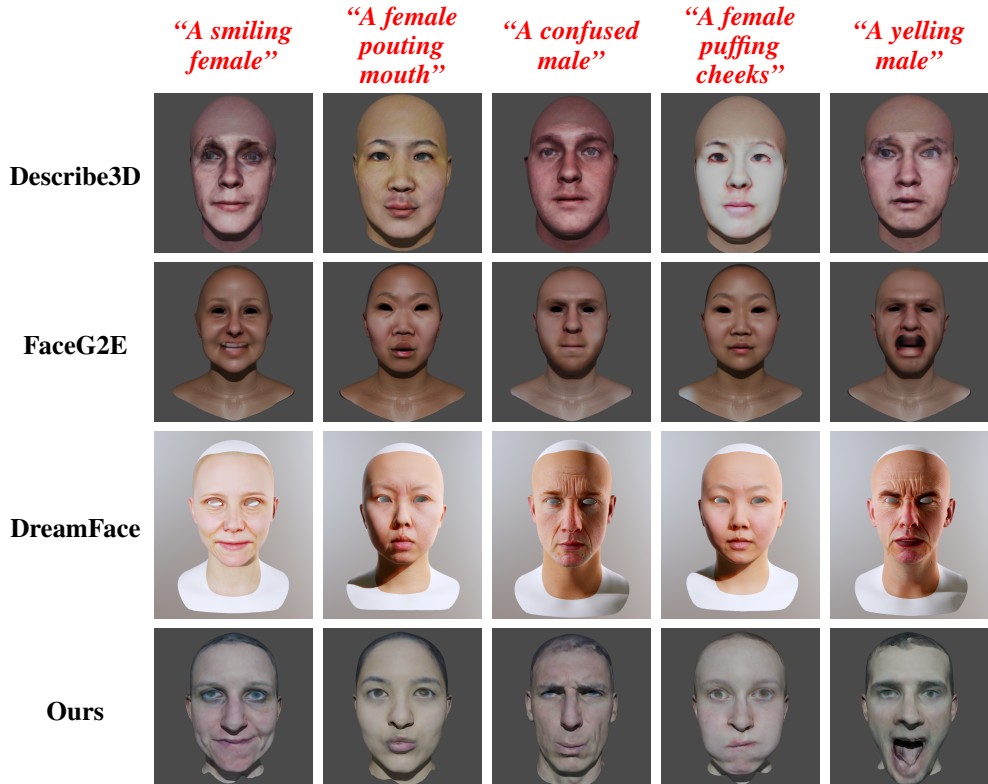

Figure 5: Generated results with different text guidance.

we adopt the strategy of selecting shorter diagonals during mesh reconstruction from the position image. Furthermore, following Gu et al. (2002), we incorporate the generated normal images to compute refined surface tangent normals for normal mapping (Cohen et al., 1998). This technique enhances the visual fidelity of the reconstructed model by simulating realistic lighting effects without modifying the underlying geometry, as illustrated in Figure 4.

## 3 DATASET

Our generative model is trained on NPHM dataset (Giebenhain et al., 2023), ontains high-quality scans of real human heads. The dataset includes over 500 individuals spanning diverse ethnicities, age groups, and skin tones, with each subject captured under 7 to 24 different facial expressions. After careful data selection and preprocessing, we obtained 7,000 textured human head meshes. To further expand the training corpus, we performed data augmentation on the corresponding geometry images. In addition to applying rigid transformations, we generated multiple conformal parameterizations for each mesh by varying the choice of corner points when prescribing target curvature. In this way, multiple sets of geometry images and textures can be produced from each head mesh. After augmentation, we obtain 118683 sets of training data. Please see more details about the dataset in Appendix A.2.

## 4 EVALUATION

We compare our method with three text-guided avatar generation methods: Describe3D (Wu et al., 2023), DreamFace (Zhang et al., 2023) and FaceG2E (Wu et al., 2024). Table 1 summarizes the number of vertices and texture resolution generated by the four methods. We evaluated the generated results both quantitatively and qualitatively.

Table 1: Comparison of generated results across four different methods, including shape representation, number of vertices, and resolutions of texture, normal, and specular maps.

| | Geometry representation | Number of vertices | Texture resolution | Normal resolution | Specular resolution |
|---|---|---|---|---|---|
| Describe3D | 3D Mesh | 26,369 | $512 \times 512$ | N/A | N/A |
| DreamFace | 3D Mesh | 14,062 | $2048 \times 2048$ | $2048 \times 2048$ | $2048 \times 2048$ |
| FaceG2E | 3D Mesh | 20,481 | $512 \times 512$ | N/A | N/A |
| **Ours** | **Geometry Image** | 260,100 | $512 \times 512$ | $1024 \times 1024$ | N/A |

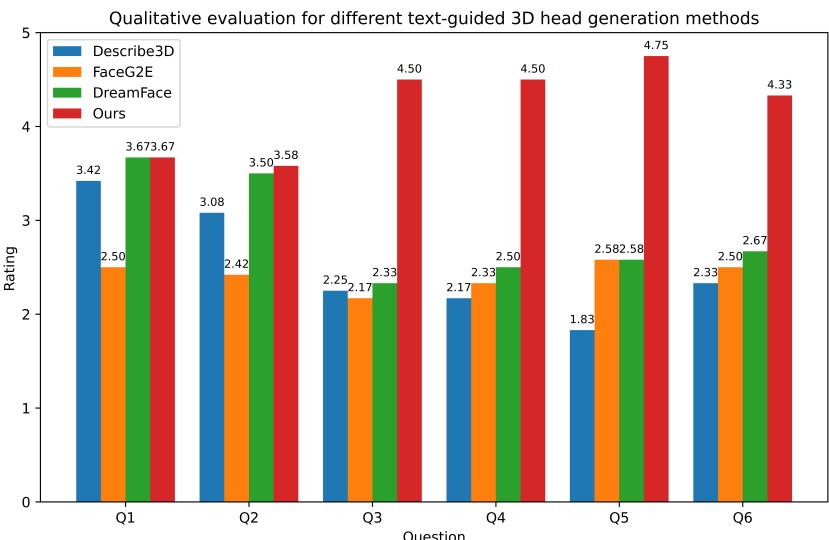

Figure 6: Results of the user study.

## 4.1 QUALITATIVE EVALUATION

Figure 5 presents rendered examples of generated 3D heads under different text prompts describing facial expressions. All results were rendered in Blender (Blender Online Community, 2025) using identical rendering settings. Our method employed the normal mapping strategy described in Section 2.5, while the DreamFace results additionally incorporated both normal and specular mapping. Among the four approaches, Describe3D is largely incapable of producing meaningful expressions. FaceG2E can synthesize simple expressions such as *smiling* or *laughing*, but it primarily modifies texture maps rather than geometry, leading to mismatches and visual artifacts. Also, limited number of vertices and triangles restricts expressiveness, and even the wireframes remain visible. DreamFace benefits from high-resolution normal, texture, and specular maps, which help preserve local details even when the underlying geometry is coarse. However, its reliance on only 14,062 vertices fundamentally constrains its expressive capacity, rendering it ineffective for complex expressions such as *pouting mouth* or *yelling*. In contrast, our method leverages the dimensionality reduction of geometry images to generate head models with substantially more vertices and triangles. This enables the synthesis of fine-grained geometric detail and greatly enhances the ability to capture complex facial expressions.

We further conducted a user study to qualitatively evaluate our method against prior approaches. The study was designed as a questionnaire comprising six questions, each rated on a five-point scale (1–5). The questions were as follows:

Q1 **Mesh quality**: How would you rate the quality of the texture-less mesh results?

Q2 **Head quality**: How would you rate the quality of the textured head results?

Q3 **Consistency (mesh and prompt)**: Do the texture-less mesh results accurately reflect the intended expressions? (Evaluated using five expressions: *smiling*, *pouting mouth*, *confused*, *puffing cheeks*, and *yelling*)

Q4 **Consistency (head and prompt)**:Do the textured head results accurately reflect the intended expressions?

Q5 **Diversity**: Does each method produce sufficiently distinct results for different text prompts?

Q6 **Overall Rating** How would you rate the overall quality of the generated results?

Figure 6 summarizes the results of the user study. Our method consistently achieved the highest scores across all evaluation criteria and demonstrated a clear advantage over competing approaches, particularly in terms of consistency, diversity, and overall quality.

## 4.2 QUANTITATIVE EVALUATION

We quantitatively evaluate each method using the **Clip score** and the **Vendi score** (Friedman & Dieng, 2023). The CLIP score measures the alignment between rendered images and their corresponding text prompts, while the Vendi score assesses the diversity of the generated outputs. To conduct this evaluation, we selected 40 prompts that specified both gender and facial expressions (e.g., "*a female pouting her mouth*") and generated 3D heads with all four methods. For each method, we rendered both textured and texture-less heads under identical rendering settings (For DreamFace we used the rendered images from their online demo[1]). CLIP and Vendi scores were then computed on the rendered images. Additionally, to further evaluate the diversity of facial expressions, we applied MediaPipe (Lugaresi et al., 2019) to extract facial landmarks from the rendered images and calculated the Vendi score on the landmark images. The quantitative results are reported in Table 2. showing that our method achieves the highest scores across all evaluation settings.

Table 2: Quantitative comparison of previous methods and ours. Clip score and Vendi score of the results, both with and without textures, were calculated. The best results are shown in **bold** and the second best are underscored. † For Dreamface only rendered images of textured heads can be obtained from their online demo.

| | Clip Score (w/o texture) | Clip Score (w/ texture) | Vendi (w/o texture) | Vendi (w/ texture) | Vendi (landmarks) |
|---|---|---|---|---|---|
| Describe3D | 23.1421 | 23.8024 | 1.8196 | 2.0289 | 1.2471 |
| DreamFace† | – | 24.0925 | – | 2.3974 | 1.4066 |
| FaceG2E | 23.1087 | 24.0103 | 1.6644 | 2.0203 | 1.427 |
| **Ours** | **23.3735** | **24.2478** | **1.9613** | **2.4849** | **1.533** |

## 5 DISCUSSION

Geometry images achieve information compression through dimensionality reduction as a form of 3D representation, enabling the expression of fine-grained 3D models with lower storage requirements. Moreover, it inherently incorporates surface parameterization and is compatible with existing computer graphics pipeline. We believe that by combining geometry images with the Vision Language Model, we can achieve more tasks beyond text-guided 3D head generation, such as 3D model editing, subdivision and super-resolution, as well as model compression.

Although the evaluation demonstrates that our method can generate high-quality 3D human head with text guidance, several limitations remain. First, despite the mitigation strategies we employed, angle distortion - particularly near image boundaries - persists as an issue. This may be addressed by calculating a quasi-conformal mapping (Zeng et al., 2009) rather than a conformal mapping or area-preserving mapping, through a trade-off between angle distortion and area distortion. Second, neural networks perform poorly on the borders of geometry images, often introducing defects along the reconstructed mesh boundaries. This could be solved by padding the image borders during training or removing the border pixels during reconstruction. Finally, our proposed method is only compatible with surfaces with one boundary and zero genus (topological disks). For surfaces with more complex topology, cuts and alternative parameterization methods would be required. Addressing these limitations will be the focus of our future work.

---

[1]https://hyper3d.ai/chatavatar

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

## A  APPENDIX

### A.1  ALGORITHM FOR OPTIMAL TRANSPORT

**Input:** A planar rectangle with measure $(\Omega, \mu)$; A point set with measure $(P, \nu)$ obtained from $\phi$; a threshold $\epsilon$.
**Output:** An measure preserving map $\psi^{-1} \circ \phi$.
$\mathbf{h} \leftarrow (|p_1|, |p_2|, \ldots, |p_k|)^T, p_i \in P$;
Compute the power diagram $D(\mathbf{h})$;
Compute the dual power Delaunay triangulation $T(\mathbf{h})$;
Compute the cell areas $\mathbf{w}(\mathbf{h}) = (w_1(\mathbf{h}), w_2(\mathbf{h}), \ldots, w_k(\mathbf{h}))^T$;
Compute $\nabla E(\mathbf{h})$ using; eq. (7);
**while** $|\nabla E| > \epsilon$ **do**
    Compute the Hessian matrix using eq. (8);
    $\lambda \leftarrow 1$;
    $\mathbf{h} \leftarrow \mathbf{h} - \lambda H^{-1} \nabla E(\mathbf{h})$;
    Compute $D(\mathbf{h}), T(\mathbf{h})$ and $\mathbf{w}(\mathbf{h})$;
    **while** $\exists w_i(\mathbf{h}) == 0$ **do**
        $\mathbf{h} \leftarrow \mathbf{h} + \lambda H^{-1} \nabla E(\mathbf{h})$;
        $\lambda \leftarrow \frac{\lambda}{2}$;
        $\mathbf{h} \leftarrow \mathbf{h} - \lambda H^{-1} \nabla E(\mathbf{h})$;
        Compute $D(\mathbf{h}), T(\mathbf{h})$ and $\mathbf{w}(\mathbf{h})$;
    **end**
    Compute $\nabla E(\mathbf{h})$ using; eq. (7);
**end**
Construct $\psi : \Omega \to P, W_i(\mathbf{h}) \to p_i, i = 1, 2, \ldots, k$.
**return** $\psi^{-1} \circ \phi$

**Algorithm 1:** Measure Preserving Parameterization

### A.2  DATASET AND DATA PRE-PROCESSING

The NPHM dataset (Giebenhain et al., 2023) provide three kinds of 3D models: the textured raw scan (a textured mesh with around 2 million vertices), the vanilla Flame (Li et al., 2017) fitting (5023 vertices without texture) and the one with subdivided face (33506 vertices without texture). The raw scan is a high-genus surface and contains hairs, shoulders, and parts of the clothing, where there are many handles. Since geometry image requires the mesh to be with zero genus and one boundary, we had to perform necessary pre-processing to meet such requirement. Specifically, we cut out the

face from the raw scan and use it to replace the face of the refined fitting model. Some necessary geometric processing algorithms and manual modifications were performed to remove remaining handles and scanning defects. Color information for areas outside the face is obtained by locating the nearest point on the raw scan and stored as vertex RGB data.

When calculating initial parameterization, We divide the mesh boundary into 8 segments based on length, then select 8 sets of vertices as corner points. So obtain 8 different parameterizations for one identical 3D mesh. See Figure 7 and Figure 8.

## A.3 Implementation Setup

The data pre-processing and OT-based geometry image generation were done on a computer with an Intel Core 14700 CPU, an NVIDIA RTX 5090 GPU and 64GB RAM. The Ricci Flow and Optimal Transport algorithm were implemented in C++. The DDPM training were performed on a server with an Intel Xeon Platinum 8468 CPU, 8 NVIDIA H100 GPUs and 1TB RAM. The training code was implemented in Python using Pytorch.

## A.4 Text Prompts Used in Evaluation

*"a smiling female", "a smiling male", "a calm male", "a surprised female", "a female with an exaggerate face", "a female with mouth pouting to the right", "a shocked male", "a male pouting mouth", "a male with closed eyes", "a male with pursed lips", "a surprised male", "a confused male", "a crying female", "an angry male", "a male with mouth pouting to the right", "a laughing male", "a sad female","a sad male", "a laughing female", "a female with pursed lips", "a female with closed eyes", "a yelling female", "a dissatisfied female", "a yelling male", "a calm female", "a male with an exaggerate face", "a painful male", "a male with puffed cheeks", "a male with mouth pouting to the left", "a crying male", "an angry female", "a confused female", "a female with mouth pouting to the left", "a female pouting mouth", "a female with puffed cheeks", "a dissatisfied male", "a painful female", "a shocked female", "a male shouting", "a female shouting"*

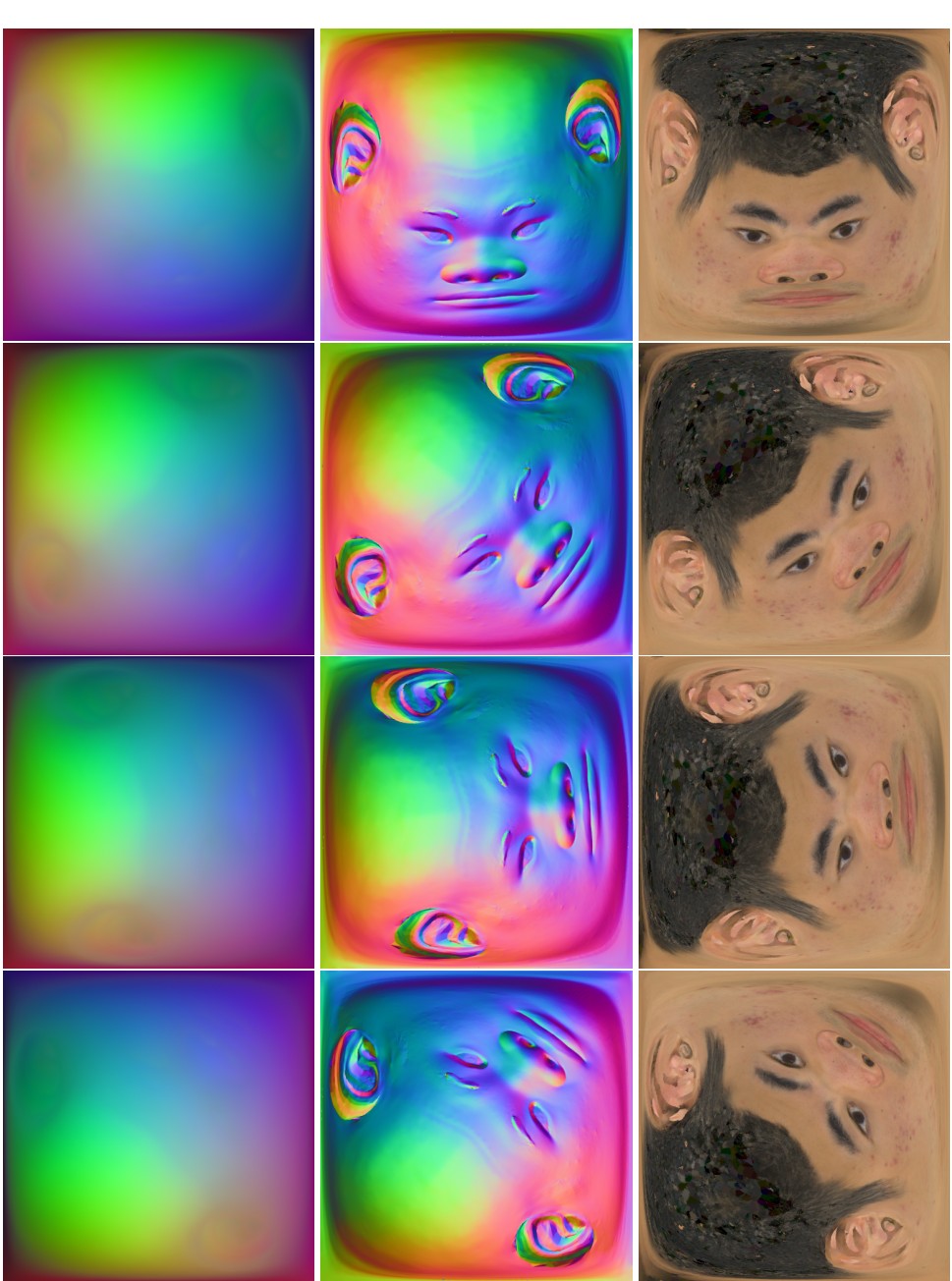

Figure 7: Data augmentation (samples 0–3).

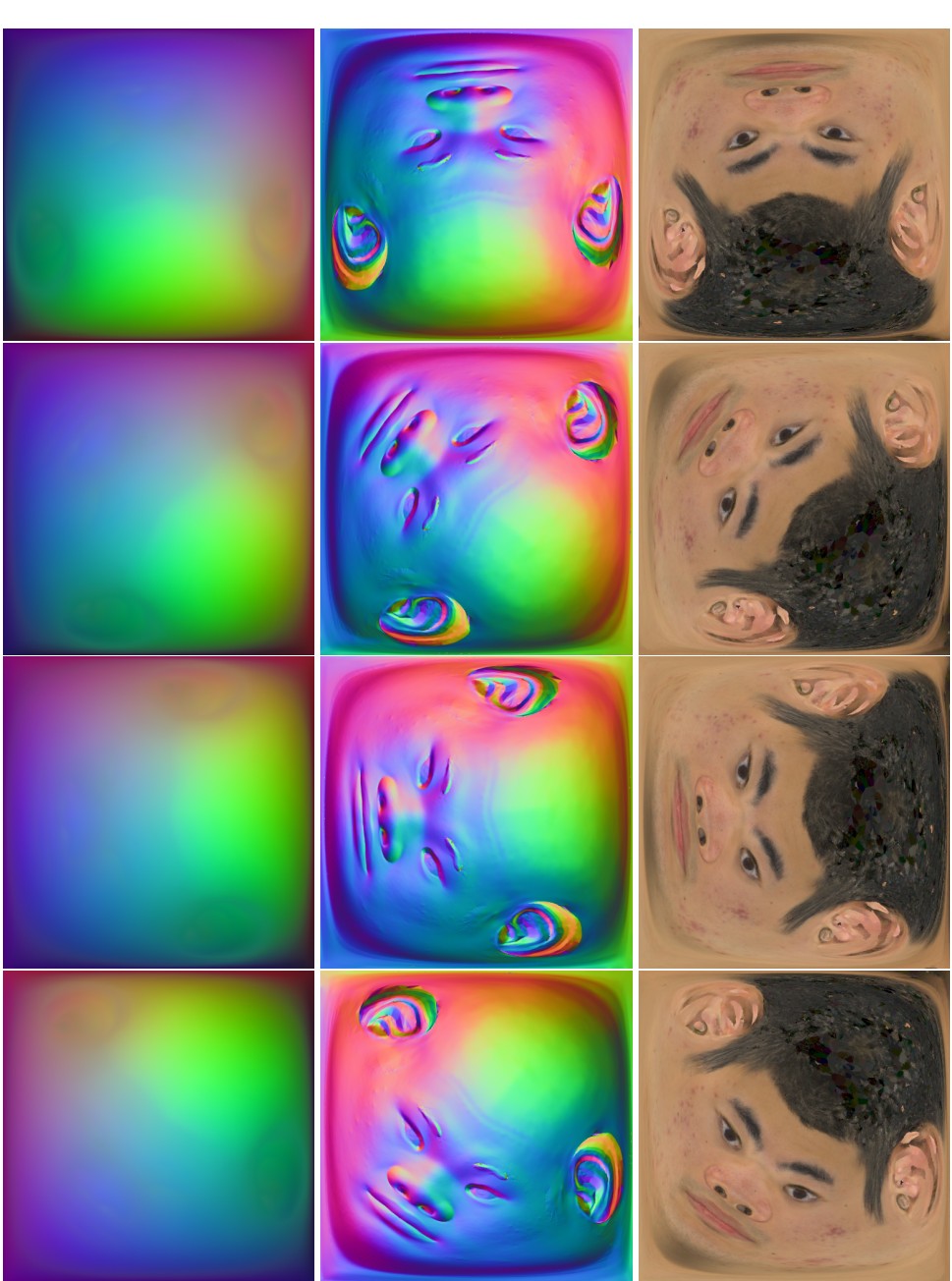

Figure 8: Data augmentation (samples 4–7).

