# OpenReview forum: "Text-Guided 3D Head Synthesis Using Geometry Images"
_ICLR.cc/2026/Conference — ICLR 2026 Conference Withdrawn Submission_

### Official Review · Reviewer_Noiq · 2025-10-28

**Soundness:** 3
**Presentation:** 2
**Contribution:** 3
**Rating:** 4
**Confidence:** 5

**Summary:**

The paper presents a framework for text-guided expressive 3D head generation that employs geometry images as the core 3D shape representation. Traditional 3DMM-based head modeling approaches are limited by a fixed and relatively small vertex count, which constrains geometric expressiveness and optimization quality. In contrast, the proposed method treats each pixel in a geometry image as an independent vertex, enabling fine-grained geometric optimization and improved reconstruction fidelity.

**Strengths:**

- The use of geometry images for representing and generating 3D head geometry is an interesting and promising direction.

- The introduction of a measure-preserving parameterization appears novel and contributes to better reconstruction accuracy, particularly in regions requiring higher detail.

**Weaknesses:**

- The paper should clarify how the proposed geometry image representation fundamentally differs from UV-based unwrapping approaches used in prior works (e.g., DreamFace), where each UV pixel also corresponds to a 3D surface point.

-  It seems that the primary improvement arises from the measure-preserving parameterization, which allocates more 2D space to detailed facial regions. Thus, the core contribution may lie in the parameterization itself rather than in the use of geometry images.

-  The advantage of having more vertices (via higher-resolution geometry images) should be evaluated more rigorously. A fairer comparison would involve subdividing the 3DMM mesh to match the vertex density and then comparing the reconstruction or generation performance.

-  The paper would benefit from a deeper analysis of the limitations of conventional 3DMM-based representations and a clearer explanation of why geometry images provide a superior alternative. If the primary novelty lies in the parameterization scheme, the paper should emphasize that aspect rather than framing the geometry image representation as the main contribution.

**Questions:**

Please see the weakness

---

### Official Review · Reviewer_5f8M · 2025-10-31

**Soundness:** 3
**Presentation:** 3
**Contribution:** 2
**Rating:** 4
**Confidence:** 4

**Summary:**

The paper addresses the task of text-guided 3D head generation, by formulating the 3D generation task as a 2D image generation task.

A core part of the paper is concerned with obtaining single-chart 2D embeddings of zero-genus 3D head shapes. Compared to existing works which often suffers from artifacts around cuts of the embedding, the proposed method exploits the fact that human head geometry (excluding hair) follows a simple topology. Therefore, they can obtain clean 2D embeddings using without any cuts.
The authors propose an algorithm to obtain measure-preserving embeddings, from which a geometry image can be obtained.

Once the dataset of (slighly simplified) 3D shapes has been embedded, the 3D generation tasks boils down to the generation of geometry images, and hence tools from text-conditioned image generation can be leveraged.

To validate the proposed method the authors set-up a user study, where the proposed method produces comparable mesh and texture quality, but significantly outperforms baselines in terms of consistency with text input and diversity. Text-consistency and diversity are further quantified using CLIP and Vendi scores, where the proposed method also outperforms all baselines.

**Strengths:**

- Casting the 3D generation problem as a 2D image generation problem is a promising approach, which enables access to a rich literature of generative methods. This approach is relatively novel for the human head domain, and object generation approaches could rely on existing UV-parameterizations.
- The proposed algorithm successfully embeds 3D objects into 2D geometry images, from which high quality 3D can be extracted again. This part has been excellently executed and provides good conditions to tackle the generation task in the geometry image domain.
- Furthermore, the ability to specify a desirable measure seems like another important addition. But to be perfectly honest the domain of conformal/measure-preserving embeddings is not in my area of expertise.

**Weaknesses:**

- Unfortunately, the paper does not provide any ablations, nor does it train baselines on the NPHM datasets. As a results, it is currently hard to understand why the user-study and quantitative evaluation favors the proposed method. The question is, is the performance gap coming from the different dataset, or is it coming from a superior method? I think this is a major issue, and there should be some hints provided to the audience in a top tier scientific conference as to what is the case, especially since the proposed method does not have a new capability that previous work could not solve.
- Another limitation of the propsed method is the restriction to zero-genus shapes. Therefore, the authors needed to simplify the full 3D scans, by taking combining the scanned facial area and registered back-of-the-head in FLAME topology. This raises the questions: How would a model perform which just uses the refined FLAME meshes, maps them to geometry images using the FLAME UV-parameterization and performs generation in this space? Would the measure preserving parameterization outperform the human crafted FLAME parameterization? Since the proposed method cannot handle arbitrary topology, I think such a comparison would have been fair.
- HeadCraft is another recent method which performs generative modelling in the FLAME UV space in order to generate mesh geometry. While HeadCraft is not text-conditioned (and works with GAN instead of Diffusion), it would be an interesting baseline, or at least it would be worth a mention.
- The formatting of math in the paper seems odd, e.g. eq. 8, the math font, line 202, 203, 216, etc.
- Generation quality: while parts of the generation results seem to be highly detailed, especially the mouth area is sometimes oddly deformed.

**Questions:**

- Typically, when diffusion is applied to rather small datasets (especially w.r.t. the number of identities), I am worried about retrieval. Namely, diffusion models are excellent in remembering training examples, and snapping to one specific sample during the diffusion process, as soon as the model is certain enough that this sample suits the text description well enough. While the Vendi score already provides some sort of measure of diversity, the score could also be good, if the model perfectly remembers the training dataset. Therefore, e.g. visualizing the nearest neighbor (e.g. as judged by chamfer distance) to each generated sample gives a good impression, whether the model can actually generated novel shapes, or whether it can just do retrieval. For quantitative metrics, e.g. the paper "HyperDiffusion" provides a small overview. So the concrete question is: To what degree does the model perform retrieval vs. generation of new shapes?
- How are text descriptions of training data obtained?
- How consistent are 2D parameterizations across different people? E.g. when ignoring the augmentations, is the mouth/eyes/noise always at the same/similar position? If not, is that an additional challenge for the generation model?

---

### Official Review · Reviewer_C1ws · 2025-11-01

**Soundness:** 2
**Presentation:** 2
**Contribution:** 2
**Rating:** 2
**Confidence:** 4

**Summary:**

The paper proposes a text-guided 3D head generator that recasts 3D mesh synthesis as 2D image generation via geometry images. It first computes a conformal (Ricci-flow) parameterization of a head mesh, then refines it to a measure-preserving parameterization (optimal transport) so sampling density concentrates on the face. The system stacks position/normal/texture geometry images and trains a conditional DDPM (CLIP text embeddings, cross-attention UNet). Generated geometry images are deterministically reconstructed into meshes, with normal mapping for detail. Trained on NPHM (7k meshes → 118,683 augmented samples), the method is compared against Describe3D, DreamFace, and FaceG2E using qualitative renders, a user study, and quantitative CLIP/Vendi scores (also Vendi on MediaPipe landmarks). Reported advantages include higher vertex counts and improved expression fidelity.

**Strengths:**

Treating head synthesis as image diffusion on geometry images exploits mature 2D diffusion tooling while avoiding multi-chart cracking issues common in prior geometry-image work; a single global parameterization suffices for disk-topology heads.

Geometry images are a strong fit for head synthesis when topology constraints hold: they reduce memory, enable high mesh tessellation at modest 2D resolution, and keep UVs aligned by construction.

**Weaknesses:**

No demo videos. Hard to evaluate the quality of the 3d synthesis and consistency of the generation.

Figure 1 generated results present explicit blurry and artifacts. I am not sure if the performance of the model is really visually acceptable or not.

The entire pipeline assumes single-boundary, genus-0 surfaces (topological disk). Hair/ears/neck/shoulders are pruned; broader head-and-hair avatars or open-world assets would require cuts/multi-charts (where prior cracking re-appears). This significantly limits applicability beyond cropped faces.

The method outputs ~260k vertices versus 14k–26k for baselines; unsurprisingly, higher tessellation helps mesh smoothness and expression creasing. A fair test would (i) upsample baseline meshes (subdivision) to matched vertex counts and/or (ii) downsample the proposed meshes to baseline counts and re-render. As presented, it’s hard to disentangle representation capacity from method quality.


Evaluation relies on CLIP score (text–image alignment) and Vendi (diversity) computed on renders; even the “geometry” variant uses landmark diversity from MediaPipe on images, not 3D errors on meshes. This leaves a gap in geometric ground-truth validation (surface distances to scans, expression-parameter accuracy, blendshape recovery). Also, DreamFace had no textureless renders (only demo images), weakening parity.

Prompts are essentially gender + expression (A.4 list), which is a narrow text space; there’s no exploration of identity attributes, fine-grained semantic controls (e.g., “wrinkled brow”), or multi-attribute composition. The “text-guided” claim is thus scoped to expression control, not general semantics.

NPHM scans were surgically edited to meet topology needs (face cut-out, replacement into a fitted template, manual modifications). While necessary, this may bias textures/geometry (e.g., hairline/ear seams, demographic artifacts) and narrows representational variety. The paper claims demographic diversity but provides no stratified performance analysis.

**Questions:**

See the weakness.

Additional questions:

Did you evaluate on unseen identities and report performance separately from identities seen during training?

Can you provide scan-referenced geometric errors and distortion metrics (not only CLIP/Vendi on renders)?

What’s the sensitivity of expression fidelity to the ROI magnification factor and to image resolution?

Beyond gender/expression, what kinds of text controls can your model follow? Any failures for fine-grained FACS-style edits?

---

### Official Review · Reviewer_FXQg · 2025-11-01

**Soundness:** 2
**Presentation:** 1
**Contribution:** 2
**Rating:** 4
**Confidence:** 3

**Summary:**

The paper presents a new framework for text-guided 3D head generation that applies the geometry images instead of the 3DMM model. To achieve this goal, the authors first compute a measure-preserving parameterization of the head meshes to construct the geometry images for the training dataset. Later, the DDPM (denoising diffusion probabilistic model) is applied to train the model for 3D head generation. Experiments on the NPHM dataset are conducted to demonstrate the performance of the method and the effectiveness of each part.

**Strengths:**

+ The use of geometry images to replace the 3DMM model is interesting and demonstrated to be useful for 3D head generation.

+

+ The method section is well-written and easy to follow.

**Weaknesses:**

- More qualitative and quantitative results and comparisons should be provided to further demonstrate the method. Rotating videos may also be beneficial to present the results.

- The presented results in Figure 5 are not good or better than existing techniques. Unwanted artifacts can still be observed. Additionally, the authors could present more results with different prompts, standard expressions, and various types of avatars, etc.

 - The ablation studies in Figure 4 and Figure 2 (last 2 rows) do not clearly demonstrate the improvement brought by normal mapping.

- The authors could provide visualizations of their processing of the NPHM dataset to help illustrate.

- There is no illustration of limitations and potential societal impacts.

- While the authors claim that the reliance on 3DMM limits the quality and generalizability of the methods, it would be better to present some demonstrating visualizations.

**Questions:**

Besides the weaknesses listed above:

+ The usage of 3DMM has another advantage: connecting the results with expression parameters and further possibly enabling the head animation. Will the proposed geometry image similarly allow for this task?

---

### Official Review · Reviewer_Zu9v · 2025-11-01

**Soundness:** 2
**Presentation:** 3
**Contribution:** 2
**Rating:** 2
**Confidence:** 4

**Summary:**

The paper proposed a method for text-guided generation of human head meshes, with an emphasis on producing meshes for a given face expression. The method is based on a diffusion model (DDPM) operating in the domain of geometry images, which constitutes a significant novelty of the proposed approach. Using geometry images allows us to abstain from using 3DMMs, since those fully define an arbitrary mesh and allow those to be generated in the 2D texture domain. Authors use NPHM dataset with augmentations to generate training data and process samples from this dataset to create geometry images for training a diffusion model. The results are evaluated via CLIP and Vendi scores & via a user study.

**Strengths:**

- Pitch of 3DMM being too low-poly for text-based generation sounds generally reasonable.
- The idea of using geometry images is elegant to be able to generate arbitrary meshes that look plausible. The description of the method is comprehensive.
- Results that are shown look generally better than the ones of the baselines in terms of matching between the text prompt and the result.
- User study shows that the distinctiveness of the outputs is much higher than that of the other methods.
- Thorough and well-described method section with many details on the geometry image generation and other parts of the pipeline.

**Weaknesses:**

- CLIP-Head not mentioned at all and not compared to (https://raipranav384.github.io/clip_head/) -- much simpler method to operate in NPHM model space. The results are quite comparable & UV creation is much easier & no diffusion model is used.
    - There might be some other missing baselines. E.g., "HeadStudio: Text to Animatable Head Avatars with 3D Gaussian Splatting" generates animatable avatars from text. Same about "HeadArtist: Text-conditioned 3D Head Generation with Self Score Distillation". "AvatarStudio: Text-driven Editing of 3D Dynamic Human Head Avatars" also solves a similar task, just in the editing regime. In "Text-Guided 3D Face Synthesis - From Generation to Editing", geometry from 3DMM is refined to match the prompt. "FaceCLIPNeRF: Text-driven 3D Face Manipulation using Deformable Neural Radiance Fields" solves the task with much higher realism than the proposed method. This is what only a rough literature search yielded; there are probably more related works.
    - Because of this, I think the pitch of heavy reliance on methods on 3DMMs is probably quite overstated. There seem to be both methods that generate arbitrary meshes, 2D images / NeRFs, and meshes produced in the implicit parametric space, like CLIP-Head. The pitch would make sense from the speed perspective (e.g., by saying that Marching Cubes is a time-consuming step), but there's a DDPM as a part of this method + can't find inference time statistics, so hard to support that too.
- Also, have the authors tried DDPM directly in the NPHM model's space? This sounds much simpler, and the only downside is the necessity of running Marching Cubes; however, running DDPM itself is a time-consuming step anyway. I can imagine that this ablation may work worse than the proposed method, but it would natural to compare to that.
- Experimental section seems to be a bit raw:
    - The only results shown are in Figure 5 -- only 5 images. Other views of these meshes are also not shown.
    - Diffusion should also allow for stochastic generation -- this seems not to be shown at all.
    - Table 2 -- improvements are very minor and not necessarily statistically significant, given that only 40 prompts have been used. The specific set of prompts could easily be handcrafted to achieve the improvement by metrics.
    - One of the main pitches of the paper, as far as I understand, seems to be about the realism of expressions. This can only be estimated by looking at many subjects in video or as many renderings under various expressions/prompts, preferably in a supplementary video or at least as a grid with visuals. It's unclear what the failure cases are (and what the quality standard for realism is that the authors consider).
        - This also raises a question of the correctness of Q1 & Q2 in the user study. The demonstrated results for DreamFace look clearly much more realistic (i.e., less uncanny), especially when zoomed in, but Ours wins over DreamFace in the user study.
- While I highly appreciate the rigour of the method section, it seems like it is more of a contribution from prior art, and the proposed method mainly seems to reuse the previously proposed approaches to generate a geometry image. I could be wrong, and there could be some improvements on that side that I missed, perhaps because it is not very clear from the text what the proposed novelty in measure-preserving parameterization is.
- Description in L647-651 about NPHM data preprocessing seems quite unclear. Same about the augmentation procedure -- it seems to yield a large increase in the dataset size, but it's unclear whether the augmentation preserves the realism of the meshes or not; should probably include some visuals to support that.
- Visuals in Fig. 5, when zoomed in on the "Ours" results, also clearly demonstrate highly uncanny artefacts (e.g., differently sized eyes, unrealistic lip shape, horror-style skin colors) -- clearly a difference with NPHM scans.
- Fig. 4: I could only observe the difference between (c) and (d) by zooming the figure in at the PDF scale of 500%. The only difference I observe is the disappearance of line-like artifacts.

**Questions:**

- What is the inference time of the proposed method?
Some typos found:
- L99 "OT-based": what's OT? Is it offline tracking?
- L362: ontains

---

### Note · Authors · 2025-11-14

I have read and agree with the venue's withdrawal policy on behalf of myself and my co-authors.